# Tracks to Postgraduate Rural Practice: Longitudinal Qualitative Follow-Up of Nursing Students Who Undertook a Rural Placement in Western Australia

**DOI:** 10.3390/ijerph20065113

**Published:** 2023-03-14

**Authors:** Caroline Crossley, Marjorie Collett, Sandra C. Thompson

**Affiliations:** Western Australian Centre for Rural Health, University of Western Australia, Geraldton, WA 6531, Australia

**Keywords:** nursing, rural, placement, student, longitudinal, experiences, career, workforce, employment

## Abstract

The nursing workforce is the backbone of healthcare provision in rural and remote Australia. Introducing student nurses to rural clinical placements is one strategy used to address the shortfall of healthcare workers outside of major cities, with the goal of improving the training, recruitment and retention of nurses in rural areas. The aim of this qualitative, longitudinal study was to better understand personal and professional decision-making around rural nursing practice intentions and subsequent rural employment and retention. The study methodology consisted of repeated semi-structured interviews with student nurses who had completed at least one rural placement and following them on their journey to becoming graduate nurses over a 6-year period. Thematic longitudinal analysis was undertaken, with three main themes developing, each with further subthemes: (1) participants’ satisfaction with rural placements; (2) their challenges with gaining employment; and (3) considerations regarding ‘going rural’ for work. The participants engaged in both prospective and retrospective reflection around several professional, personal, and wider systemic barriers and enablers to rural practice, which are discussed in detail in this paper. The insights from this longitudinal study have the potential to assist the development of a sustainable rural nursing workforce through informing rural workforce programs, strategies and policies.

## 1. Introduction

Improving the rural health workforce is of paramount importance in sustaining good health outcomes in communities throughout rural Australia. Rural communities suffer disproportionately with higher rates of chronic disease due to the social determinants of health, ref. [1] meaning that rural communities require access to equitable and sustainable healthcare. Nurses make up the largest healthcare profession within rural and remote Australia [2]. However, the figures required for a sustainable nursing workforce look bleak, with the prediction from Health Workforce Australia 2014 indicating a shortfall of nurses expected to be in the region of 85,000 by 2025, worsening to 123,000 by 2030 [3]. With demand expected to exceed supply and domestic attrition rates recorded at 34% in 2012, there is a requirement for robust nursing workforce planning. Given the reliance of the rural health workforce on nurses, it is particularly important that efforts to address and avert this overall shortfall include the training of nurses that have both the desire and capability to work in rural health settings.

In recognition of the health workforce needs in rural areas, the Australian Government has established University Departments for Rural Health (UDRHs) in rural sites across Australia [4]. A core aim of the UDRHs is to develop the rural health workforce to meet the healthcare needs of rural communities whilst developing strategies to promote the rural and remote health workforce [5]. While different definitions of rural are used in different countries, in Western Australia rural is used to refer to areas outside of the metropolitan area, with classifications generally based upon issues of remoteness, ease and regularity of access, and population size. Well-supported rural clinical placements for nursing and allied health students through the UDRHs is a strategy used to increase interest in rural health careers and to increase the likelihood of graduates entering the rural workforce [6,7,8]. In addition, a rural background and previous rural living are known to be factors that increase the likelihood of later working rurally [8,9,10,11,12,13].

Currently, Australian nursing students are required to complete a minimum of 20 weeks or 800 h of quality professional experience placement to qualify for registration with the Australian Health Practitioner Regulation Agency (AHPRA) [14]. However, rural placements are not currently mandatory. At the time this study commenced, the Western Australian Centre for Rural Health (WACRH) was the only UDRH in Western Australia (WA) and supported nursing students from various WA universities, which differed in their admission requirements and philosophy, including their approach to rural training.

Detail about the roles and merits of nursing clinical preceptors and clinical facilitators is available elsewhere [15,16,17]. In summary, during clinical placements, students are assisted in acquiring the necessary nursing skills, linking theory to practice, and meeting the required nursing competencies by preceptors and clinical facilitators. A preceptor is an experienced registered nurse (RN) who supervises nursing students during their clinical rotation and is accountable and responsible for day-to-day oversight in the clinical environment and for the formal assessment of the required clinical competencies, including demonstrable attainment of the seven Nursing and Midwifery Board Australia (NMBA) Standards [14]. A clinical facilitator (CF) is a registered nurse involved in current nursing practice who is engaged to facilitate student learning both on-site and remotely. The CF role has much less face-to-face oversight in the clinical environment, having a more pivotal role liaising between the university, placement site, and students. The CF helps the student become familiar with the objectives, assessment processes, and the nature of the clinical experience prior to their commencement of the practicum. During regular mandatory meetings, the CF may advocate on behalf of the student and support the facility if problems arise regarding performance.

Research, to date, indicates that there are multiple influences on a nurse’s choice of work location. A recent systematic review [18] found that there were three key dimensions to this decision-making, including personal factors, professional factors, and the place itself, each with further subdomains of influence. They identified a gap in the knowledge around the interplay between these domains and recommended further research in this area. In particular, they recommended the use of ‘autobiographical (or life history)’ approaches to better understand ‘the flow of their lives through time and space, and the interrelation between the two’ as well as ‘to uncover nurses’ rationale for their career decisions in the context of the relationships with people and places in which they were made’. Other prior research has more specifically focused on the influence of student placements on rural recruitment and retention. Terry’s model of ‘Rural Nursing Workforce Hierarchy of Needs’ showed that clinical-related factors were ranked most highly by students in their decision to undertake rural practice after graduating [19]. These included patient safety, positive relationships among nursing generations, supportive working environments, job satisfaction, autonomy, and respect. Other studies have indicated the importance of the social factors related to rural student placements and becoming part of the community [20,21,22].

The aim of this qualitative, longitudinal study was to address some of the above gaps in the current literature, as well as to build on current knowledge to further establish to what extent a student nurse’s clinical rural placement, choice of course or university, level of support, employment opportunities, and other personal and professional factors inform or influence the graduate’s career pathway, as well as how these factors may interact. Our longitudinal and inclusive approach aimed to allow for reflection on those factors which are considered important in the decision-making process, both prospectively and in retrospect. It is hoped that this will better illuminate factors which act as barriers or enablers to the initial decision to work rurally as well as the decision to stay, and how these factors may change over time. It is also hoped to close the knowledge gap of how rural student placements interact with other personal and professional factors to influence rural work post-graduation. Closing the knowledge gap in this field will enhance the development of a sustainable rural nursing workforce through informing rural workforce programs, strategies and policies for the recruitment and retention of rural nurses.

## 2. Materials and Methods

This qualitative, longitudinal study used semi-structured interviews and was undertaken over a 6-year period from 2015–2021. Students from four universities took part in the study consisting of three metropolitan-based universities (with clinical placements facilitated by WACRH) and one eastern state university (facilitated by local university staff). A career trajectory was followed from completion of a rural clinical placement in WA to qualification and beyond.

### 2.1. Ethical Considerations

Ethics approval was provided by the University of Western Australia (UWA) Human Research Ethics Committee in 2015 (RA/4/1/7231). The researchers were based in the Midwest of Western Australia (WA). The relevant areas of the nursing students’ enrolling universities agreed to student participation in the study. University consent was gained via email, highlighting evidence of UWA Ethics approval and an outline of the student’s involvement and relevant documentation.

Relevant information was shared with individual participants about the study, including the purpose and any possible risks, and consent was obtained. Participants were all made aware that the interviews were to be audio-recorded if they consented.

Interviews were conducted by research staff who were not involved in the student’s clinical placement supervision or assessment. Students were reassured that their decision regarding involvement in the study would have no bearing on academic results, and the baseline enrolment only proceeded after unit assessments had been completed, at least two weeks following the completion of their practicum.

### 2.2. Data Collection and Analysis

Participants were selected for the study using purposive sampling. Following a rural clinical placement, nursing students were invited via email to participate in the study. Each participant was phoned individually, with the first interview occurring shortly after their rural student placement. The interviews were conducted in a one-to-one telephone format by academic nursing and medical staff based at WACRH. The interviews began with the collection of demographic data such as degree course, age, marital status and whether they had dependents. The interviewers then used pre-designed interview checklists to ask both defined questions and additional questions emerging during the conversation. The checklists differed by interview number and were designed by academic and clinical nursing and medical staff. They can be obtained by interested researchers by emailing the authors. Interviews varied in length, depending on participant time and response engagement, and generally lasted up to an hour.

At the time of the initial interview, some participants agreed to further follow-up. Participants were followed up initially by an email requesting a further interview. This was followed by a second email if there was no response. If there was no response to the second email, the participant was phoned; if participants were not able to be contacted or they declined a further interview, no further contact attempt was made. Given our interest in longitudinal reflections on their training and career, this analysis focuses only on those participants for whom at least one follow-up interview was available. There were a maximum of three interviews per participant, over a six-year period, between 2015 and 2021. Of the twenty-three participants who completed an initial interview, ten completed at least two interviews, with seven participants completing all three interviews. Thus, only participants who had completed either two or three interviews were included in this analysis.

Responses to the first round of interviews were recorded using field notes at the time of the interview, and subsequent interviews in the second and third rounds were audio-recorded and transcribed. Qualitative grounded theory with inductive reasoning underpinned our analysis. This involved systematic immersion in the data through multiple readings, detailed coding, and identification of emerging themes by three researchers [23]. Collaboration and communication within the research team were used to refine and reduce the identified themes, a process that strengthened the trustworthiness and credibility of the findings [24,25]. The saturation of the responses appeared to be reached by the time of the final interviews, with emergent patterns and subsequent themes identified. Data were triangulated over time, with interviewers accessing prior interviews of the participant before the completion of the second or third interview. This enabled clarification of previously ascertained information as required, as well as participant feedback. Investigator triangulation was also utilised to minimise the risk of bias during the analysis of the data.

For the purposes of quick identification, where quotes are used in the analysis, the notation is written Participant_University_Interview number_Year of a university course at the time of the interview (UC) or years post-graduation at the time of interview (PG). For example, for a quote from Participant A’s first interview in their first year of study, the quote is annotated A_UWA_I1_UC1. Where low-population locations were mentioned by participants within these quotes, the location is concealed for anonymity purposes and is written as <location*> in the article text.

## 3. Results

### 3.1. Participant and Course Information

All participants were female, with ages at the first interview ranging from 19 to 48 years. There were four participants who attended the University of Notre Dame (UND) and two for each of the University of Western Australia (UWA), Edith Cowan University (ECU), and the University of Southern Queensland (USQ) (Table 1). The ECU and UND courses are both three-year undergraduate courses based in Western Australia. The ECU course can be completed partially online. In the context of our study, USQ has an online nursing degree course that can be completed by students who are undergoing their nursing training at a regional university centre: students complete their learning both online and in-person with localised support. The UWA course was a Masters level two-year degree and has since been discontinued.

#### Summary of Work Locations at Final Interview

Three of the participants (B, D, and G) were rurally based at the beginning of the study (one in the Southwest and two in the Midwest regions of WA). All three remained rural at the end of the study. Of the seven participants (A, C, E, F, H, I, and J) based in the metropolitan area at the beginning of the study, two had moved into metropolitan-based fly-in fly-out rural roles by their final interviews (E, H). One other participant had worked rurally for some time after graduating but had relocated back to the metropolitan area by the end of the study (F). Of the five participants working solely in the metropolitan area at the end of the study, three said they would consider working rurally if their circumstances were different, one commenting that she would definitely be rurally based if not for her enrolment in further city-based study. There was one participant who commented that she would not consider working rurally unless it was a last resort (I).

### 3.2. Thematic Analysis

There were three main themes from the longitudinal analysis: (1) the participants’ satisfaction with rural placements; (2) their challenges with employment in general, and (3) their considerations regarding ‘going rural’ for work. Several subthemes were identified within each of these areas, as outlined below.

#### 3.2.1. Satisfaction with Rural Placements

The ‘Satisfaction with rural placements’ theme had five identified subthemes: A. Feeling valued; B. Support mechanisms; C. Professional skill development; D. Personal skill development; E. Considerations of rural placements.

A.Feeling valued

Participants spoke of the benefit they derived from feeling valued as part of the team and community during their rural placement. Many identified that the rural culture and mentality were different to that of metropolitan hospital placements, and they frequently spoke of the welcoming, friendly, and supportive atmosphere.


*‘I wanted to stay way after my shift finished… real community feel… I really wanted to join in’.*
(A_UWA_I2_PG6)

Participants felt that on-site RN preceptors seemed to trust and ‘believe in’ them to complete tasks more autonomously and independently while still providing support. They also spoke of the less hierarchical structures in place in the rural setting. Participants valued rural nurses having more time for them and taking the time to teach them.


*‘Just given the chance… just having someone believe in your ability made you feel confident… I think that sort of is a country mentality ’cause you found that more there than you do in the city’.*
(A_UWA_I1_UC3)

The students also appreciated their inclusion in activities, being able to get to know people in the community, including fellow staff and students, and being invited to social events by hospital staff.

B.Support mechanisms

Mixed views were expressed about the support of onsite RN preceptors while on rural placement. On the one hand, some spoke of ample opportunities to attempt many different skills with support, being encouraged to ‘have a go’ and their preceptors being more interested than they had previously experienced in providing them with feedback. This support helped them overcome the initial fears that they had regarding working in a rural site and was an affirming and valuable experience, as described by this participant: *‘I went up there by myself, not knowing anyone and was so heavily supported up there and made friends that I’ve still got now’* (E_UND_I3_PG4). They also valued being able to contact both their home university as well as local support if needed. However, a couple of participants described difficulties that reflected the small rural setting where they were placed and the clinical load. For example, two reported experiencing ‘quiet’ times during placement, with on-site staff reportedly unable to support their move to another area of the hospital. This was felt to be a lost opportunity for engaging in a deeper learning experience through work in another area of their interest. Discontinuity of preceptors was also a recurring issue, as it made it difficult for students to complete assessment paperwork. Difficulties with computer access were experienced by some, and this also impeded their feeling supported and impacted remote placement facilitation.

C.Professional skill development

Rural placements were reported to expose participants to more holistic and diverse skill development compared to being in a specialised tertiary setting. *‘You can get caught up in big hospitals and looking at patients as parts, the biomedical model… in country hospitals you just don’t, it’s different’* (C_UND_I3_PG6). Their learning included being trusted to perform procedures, understanding team dynamics, and being taught communication skills. They described how nurses generally had time to teach and assess them and had a high level of knowledge. Students commented that while on rural placements, they were better able to perform and consolidate their skills rather than just watching.


*‘I just found that I learned significantly more on my rural placements and just saw things that you would never see in the metro area… it is a lot of general stuff which is great because you get to see a wide variety of things’.*
(E_UND_I2_PG1)

Students who subsequently started working as nurses in metropolitan settings also spoke of the long-term benefits of having gained an understanding of how people from rural areas have different health needs, perspectives, and priorities, including their improved understanding of Aboriginal culture.


*‘I feel a more connected way of looking at things… why they want to get back, what’s important to them, they want to be back with their family, they don’t want to be in hospital away from everybody… the drive for them to get better, the quietness and the reluctance to talk about it because they don’t want you to keep them in there longer’.*
(A_UWA_I2_PG6)

Participants described being able to see how staff communicated with patients in rural settings as beneficial, helping them learn to build relationships with patients themselves.

D.Personal skill development

Participants spoke of the personal changes and breakthroughs that came from their rural placement experience, including enhanced cultural understanding, feeling more self-assured after living in a different place, and increased confidence that they could work rurally.


*‘It helps your development as a professional, but it also helps you personally as well – it makes you a more rounded person, different experiences outside of what you’re used to… I never even considered rural until I went on student placement’.*
(C_UND_I3_PG6)

Some spoke of their experience of living in a shared house during their placement, in which they learnt more about themselves and others. Some younger participants described their placement as ‘pivotal’ and life changing. Living on their own for the first time necessitated independence away from their usual friends and family. They learnt to be proactive and organised, with enhanced confidence to try new things.


*‘Being away from home and being away from my family… that was really the first time I was away from home by myself for an extended period, so I felt it was good in terms of independence and learning about the working lifestyle… It’s made me think there’s a lot out there to see and learn, take on any opportunity—you have to just go and give it a try whether it be in the metro or rural area… it definitely made me more confident in trying different things’.*
(F_UND_I3_PG5)

E.Considerations of rural placements

Financial considerations of rural placements were mentioned by several participants. Accommodation being organised and free was a valued aspect, particularly as some could not work while they were away and were still paying the costs of their usual accommodation back home.


*‘Students can’t afford to just go somewhere else and not be working for that period and then have to pay for accommodation there as well’.*
(J_UND_I2_PG2)

Other key considerations were being away from family for a prolonged period, particularly for those with children, the difficulty of driving for long distances alone, and the benefits of living with other students while away from home. Some participants expressed the view that rural placements should be a requirement for all nursing courses.


*‘I think it should be mandatory to do rural placements—(university) makes it so easy—free accommodation, really supportive, you’ve got someone there supporting you or just a phone call away’.*
(E_UND_I2_PG1)

#### 3.2.2. Challenges of Employment over Time

A.Graduate programs

Most participants spoke of the difficulties of gaining employment as a nurse after graduation, particularly the difficulty and competition around gaining a ‘graduate program’. All the participants were eventually offered graduate programs, but only one was successful immediately after graduation, with the rest ranging from second-round offers to one being successful only on her sixth application. Two nurses turned down their eventual offer of a graduate program in favour of roles they had gained in the meantime. Participants spoke of the importance of being known to staff at the site at which they applied, with concerns about being less likely to gain a graduate program at a chosen site if they had not been able to secure a student placement there. Following graduation from university, one nurse later considered herself ‘stuck between a rock and a hard place’ as she was still trying to successfully gain a graduate program, but because she had been employed as a casual during that time, she was no longer considered a new graduate, making it increasingly difficult. Another spoke of how she found it very hard to turn down an offer of a graduate program due to the ‘mentality and pressure’ around it, but she did so as she had been offered a community nursing job in an area of her interest.

Participants spoke of the competitive nature of the graduate program application system and the complexities of placing rural and regional sites as their preferred location choices, particularly due to being unlikely to be offered anything other than their first choice. With limited places available in regional and rural centres, putting these centres as a first choice was seen as risky due to being unlikely to be offered a tertiary hospital program as a second choice. Similarly, it seemed that placing regional or rural hospitals as a second or third choice was unlikely to be fruitful.


*‘When I was applying for grad connect you had three preferences and in the first round the applications get sent off to your first preference. And then theoretically anyone who doesn’t get a place in first round goes to second round, but in practice there is so much more applications than there are spaces that it fills up in the first round and that’s all there is… So for that reason I didn’t apply to a rural or regional hospital because the majority of them have two places or maybe eight places, it seemed like too much of a crapshoot’.*
(J_UND_I2_PG2)

One participant reported that her university encouraged students only to apply for tertiary hospital graduate programs, and she was unaware of the options available to her in regional and rural locations. Retrospectively, she felt this was an issue, but at the time of the decision, she was determined to be placed at a large metropolitan hospital.

Three participants completed their graduate programs in a regional or rural area. Of these, two had later concerns that they were lacking certain clinical skills and spoke of the inconsistent support that they had received, offset somewhat by the support they felt from other graduates and senior nurses. They suggested that opportunities for rural nurses to work within metropolitan hospitals for short periods of time would allow for more learning within a tertiary teaching environment, in particular, understanding plans for patients who have returned to them from the metropolitan hospitals for ongoing care.

B.Insecurity of employment

Participants spoke of their difficulties gaining employment due to the competitiveness of both rural and metropolitan jobs. After the completion of their graduate program, many spoke of the difficulties of gaining permanent employment, with all the nurses interviewed after their graduate program being on casual, temporary, or short-term rolling contracts immediately after completion. Many had to join a casual pool at the same hospital where they had completed their graduate program.


*‘Just playing it by ear and see what opportunities come up and go from there. It’s difficult to get permanent contracts in the public sector’.*
(F_UND_I3_PG5)

Only three nurses had been offered permanent roles by the time of their final interviews. Many had a preferred employment location but had to work elsewhere, taking any job they could find. For example, a nurse who had obtained a part-time permanent role described not really wanting to stay in that role, but she considered herself lucky to have the position.


*’I meet a lot of senior nurses… they all struggle to get permanent jobs…and they are senior and I’m junior, still better than nothing, I’ll hang on to it’.*
(I_UWA_I2_PG6)

Two of the participants spoke of how they had gained employment rurally through networking and phone calls rather than through more formal channels. For example, one participant gained employment in the rural area where she had been placed as a student by phoning the nurse manager, who remembered her and immediately offered her employment. Another commented, *‘If it’s something you want to do, it’s something you have to do yourself and be quite proactive. I opened myself up to opportunities’* (H_ECU_I3_PG3). Others described their efforts to continually upskill and network in the hope of eventually obtaining a job in an area of interest.

#### 3.2.3. Considerations regarding ‘Going Rural’ for Work

A.Family considerations

Some participants expressed their desire to work rurally but found that they were unable to do so due to family commitments, such as partners (five participants), children (three participants) and other family members (three participants) being based in a metropolitan area for work or schooling. Another spoke of how she is not currently able to complete the upskilling that she would need to be able to work more rurally due to childcare commitments.


*‘I think maybe if I was single and I’d moved out there by myself I would have been fine with the lifestyle but when you have attachments in Perth it’s not so easy… I did enjoy working rurally but what I realised is that it doesn’t quite fit with my life’.*
(J_UND_I2_PG2)

One participant reported contributing to the rural nursing workforce through a fly-in-fly-out rural role despite the challenge related to her metropolitan-based partner.


*‘I’ve tried to get my partner to move regionally but he won’t, so the next best thing is to have a job where I’m still based in Perth so it’s really kind of the best of both worlds’.*
(H_ECU_I3_PG3)

Another who was interested in rural working had wanted a graduate program in the city, as she felt that she needed to be around family in her first year of nursing for support.


*‘To do my first year in metro and to have the support of my family nearby, I’m very family orientated. Doing rural as my very first year as a grad I think would have been very challenging, but that’s not to say that I won’t now take a job in a rural area’.*
(E_UND_I3_PG4)

Another participant commented that rural working would compromise certain aspects of her lifestyle, such as specialist sports and activities.


*‘I like rural nursing… but it’s a big sacrifice to give up my lifestyle and family’.*
(I_UWA_I2_PG6)

The three participants who completed their studies in regional areas all remained working in regional or rural areas for similar reasons, including family needs affecting their options.


*‘I feel that we do lack here being able to improve our learning in a tertiary hospital, because we’ve obviously got limited facilities here and the hospital is so short staffed. But I live here with my family…and we don’t want to live in the city and I don’t think at this stage we would go any further rural until my child was at boarding school or finished high school’.*
(B_USQ_I3_PG3)

B.Skill requirements to work rurally

Many participants spoke of feeling underqualified for rural positions and the need to work in the city initially to upskill.


*‘It seems like rural nurses need more experience and more skills so it seemed if you work in the metro you have the support and learning development, and they can take you on with little experience’.*
(J_UND_I3_PG3)

Participants’ views of the skills needed to work rurally changed over time. For example, the three nurses who had trained and only worked rurally in their first interviews spoke of how they felt working and completing a graduate program rurally would be beneficial due to the greater variety of presentations and country nurses being ‘good educators’. For example, *‘A lot of general nursing, not having to specialise too much too early on…I feel like it will give a lot more rounded experiences’* (G_ECU_I2_PG1). However, by the final interview and after the completion of their graduate program, two spoke of how they felt their professional development would be enhanced by being able to further upskill at one of the tertiary hospitals in the city. They were not confident that such a secondment would be possible, particularly considering the staff shortages at their rural hospital.


*‘I have asked when I had my professional development interview whether there’s any possibility of doing a three- or four-week stint in Perth to do like a supernumery to see how they do things and to upskill that way but I haven’t heard anything from that… at the moment they don’t have the staff’.*
(B_USQ_I3_PG3)

Although both initially spoke of the possibility of working more remotely, in their third interviews both also felt that they needed more experience before being confident enough to do so.


*‘[Working rurally] that was my big long-term goal, but I think in nursing to work rural and remote you actually need to be really experienced and have quite a lot of clinical skills under your belt… so I pretty much need to keep going. Now I’m going for the next five years and see what happens… and I think the fact that you’re limited without having any tertiary experience becomes more apparent as you become more experienced’.*
(D_USQ_I_PG3)

One nurse who trained at a metropolitan campus took a rural position just after graduating (because she could not find any other work) and described it as being ‘*thrown in the deep end*’, although she felt that she had been well supported by other staff. Another nurse working in the city at the time of her final interview commented that she would have liked to gain more advanced skills as a student so that she felt able to go rural sooner after graduation.

C.Influence of rural placement

Participants spoke of the positive impacts of their rural placement on their interest in rural work, such as understanding the diversity of roles in rural hospitals, enjoying the feel of the community and the quality of life, and of the respect and time given to them when they were students. For some, their rural placement consolidated that rural work was something that they aspire to do, despite their better understanding of the challenges of that environment. Some described that a rural placement enhanced their confidence to apply for and accept rural roles after qualification, thereby expanding the range of career options available to them.


*‘That prac definitely influenced and reassured me that I can be away for a period of time and make friends and be well supported… integrating yourself into a community and understanding what that place is about is really nice… and all the regional pracs that I did assisted with this decision to do rural and appreciating and understanding it a bit better. Even the skills I learned up there… those skills I’ve retained and use now’. ‘If you have that understanding and that empowerment from that very early stage of my nursing career it sets you up to know that you can move away and it will be ok’.*
(E_UND_I3_PG4—nurse now working in a rural fly-in-fly-out role)


*‘Before my grad program I did those couple of jobs in the country… Yeah, I just felt confident in living away from home to work because I’d had those experiences in <location*>, I just feel like in that way it increased my confidence to go out there and give things a go in the country’.*
(F_UND_I3_PG5)

For some, their rural placement had more profound effects on their career path. They described how their rural placement reassured them that rural work would be a good opportunity and something that they were capable of with support systems in place.


*‘I never even considered rural until I went on my student placement’. ‘That initial placement in <location*> just really just opened things up for me, it really changed my life it really did…it really helped shape me as a person and as a professional. All up I’ve gone to <location*> including being a student… I’ve gone back four times…and that’s all from just that placement’. ‘I just found the lifestyle so nice, I loved working for a small country hospital, everyone knew everyone, I loved getting to know the patients, to walk to work, everyone was so friendly, I made so many friends that I’m still friends with today… just a really nice lifestyle, that’s why I keep coming back’. (Nurse that went back to work in rural placement town after graduating)’.*
(C_UND_I3_PG6)

Other insights were gained, as in the case of a nurse who had noticed on her placement the transient nature of rural workers. She found knowing that she did not have to commit to long-term rural work if it was not for her made it more attractive. 


*‘It’s kind of nice knowing that you don’t have to commit to staying rurally if you don’t find that it is what you want but you can fully embrace the time that you’re there and still get a lot out of it’.*
(A_UWA_I2_PG6)

Those nurses who had completed their training and graduate program in the same rural setting spoke of how it was more comfortable and less daunting to complete their graduate program in a place where they already knew the ward and staff. *‘Knowing what the ward was like, and I did get a grad, it wasn’t so daunting, and already being familiar with people there was a good thing from having done my pracs there’.* (B_USQ_I3_PG3).

## 4. Discussion

Rural placements are one of several approaches implemented as part of the long-term strategy to address the shortage of healthcare workers outside of major cities and ultimately represent one component of the efforts to overcome the healthcare disparities that exist between rural and metropolitan populations. This approach aims to improve the training, recruitment and retention of health practitioners in rural areas. However, there is relatively little in the literature beyond the immediate impacts of rural student nursing placements, particularly exploring the mechanism by which rural placements influence rural practice intentions and the subsequent transition into rural employment. This requires detailed follow-up of students and graduates over time to truly understand their thoughts and actions at each critical stage of their nursing career. This long-term perspective is particularly important given that recruitment to rural areas does not automatically lead to retention, as is made evident by the high turnover seen within the rural nursing workforce.

### 4.1. Satisfaction with Rural Placements

Well-supported, well-planned, and appropriately funded rural placements have been shown by multiple studies to increase the rural practice intentions of health professionals and, in some cases, improved graduate rural employment [13,21,26,27,28,29,30,31]. For example, medical, nursing, and allied health students from across Australia who were satisfied with their rural placement were more likely to have a rural practice intention [21,32], with placement satisfaction having an impact on changing student practice intentions from negative to positive [33]. However, there are conflicting views and evidence on exactly how placements complement prior rural intentions and rural backgrounds in encouraging resultant rural practice.

In our study, participants described feeling welcomed, valued and supported while on their rural placement. They identified the professional and social benefits of feeling part of the healthcare team as well as the wider community. This is an important aspect of rural clinical placement as health student satisfaction and rural practice intentions are linked to interactions with a ‘positive, friendly and supportive’ working environment and community engagement opportunities, a term referred to as ‘ruralisation’ [6,17,28,34]. The importance of rural placement models that include a focus on socialising students into rural communities to further engage them has also previously been noted [20,21,22].

Participants in our study particularly valued the autonomy and independence that their rural placement allowed them, as well as the development of a diverse clinical skill set, which was developed while feeling well supported by both RN preceptors and other students. They also valued positive structural factors, including free accommodation and the social aspect that this entailed. Terry’s model of ‘Rural Nursing Workforce Hierarchy of Needs’ showed that clinical-related factors were ranked most highly by students in their decision to undertake rural practice after graduating [19]. These included patient safety, positive relationships among nursing generations, supportive working environments, job satisfaction, autonomy, and respect. These factors are similar to those identified by our participants. This indicates that clinical facilitators and preceptors that focus on these factors during a rural placement experience could potentially enhance both placement satisfaction and the subsequent intention to work rurally. Therefore, promoting these factors during rural placements may be of benefit in influencing potential rural nurses of the future.

The two main negative issues described by students with respect to rural placements were quiet periods and the discontinuity of preceptors. The ability to move students to other areas is dependent on the individual preceptors on-site, including their workload and preferences for student allocation. This made it difficult for some students to have adequate exposure in smaller hospitals, even if clinical facilitators advocated for such movements. Preceptor discontinuity is a well-known issue that has previously been mentioned in several studies. Collett and colleagues noted that the capacity of nurses to undertake preceptor roles was a concern, especially due to staff shortages and high staff turnover in some areas [35]. Some preceptors were not well prepared for the completion of student paperwork or lacked the time or skills to do so. Preceptors have also reported concerns about not receiving adequate information about student requirements from the university, the challenges of their own workload, time constraints, a lack of support and recognition for mentoring students, a lack of information about the student’s needs, and difficulty accessing their own continuing education [16,36,37]. Improved communication and the provision of information about students attending rural clinical placements are important issues to address given that negative rural placement experiences have the potential to negatively impact student placement satisfaction and, hence, reduce the positive impact of rural placement on rural practice intention.

It should also be noted that the longitudinal nature of this study highlighted the value and satisfaction of the rural placement experience, even for those who did not take up rural work. In their follow-up interviews, participants reflected on the benefit they derived from understanding the differing health needs, perspectives and priorities of patients from rural areas, even when working in metropolitan areas.

### 4.2. Decision-Making: Influences on Rural Practice

#### 4.2.1. Influence of Rural Placement on Rural Practice

As graduate nurses, the participants looked back on their rural placement and spoke of the community feel, quality of life and the respect and time given to them by rural nursing staff as encouraging drawcards for rural practice. This is important because a sense of belonging or connectedness to rural life through previous exposure has been shown to influence the attraction of nurses to rural areas [18]. Similar to studies of allied health remote placements, including in the same region, this study also noted the longer-term effects of a rural undergraduate placement on personal and professional growth, in particular, the effect that it had in enhancing confidence about being able to cope with rural practice [38,39]. Participants in our study spoke of their enhanced confidence to move away from home, ‘give things a go’, and meet new people, along with the positive impact these had in applying for and accepting rural roles.

Others spoke of how the placement had consolidated their rural intention, despite understanding more of the challenges. Experiencing the challenges of rural practice as part of the rural placement experience is therefore not necessarily a negative; rather, it can potentially encourage people who will be more likely to stay in rural areas because they understand in advance the nature of the role and the support systems that they require. Similarly, the retention of allied health workers has been shown to be influenced by understanding the challenges of rural practice prior to practising there [40].

Of those participants who had trained entirely rurally, all of them remained rurally employed at the end of the study. Among factors influencing their location choice (as described below), they explained that being familiar with the location and staff meant it was less daunting to continue practising rurally following graduation. They also had chosen their course because it meant that they could train rurally and expressed no intention to leave for the city or other areas once they graduated. This shows the positive implications of encouraging and enabling healthcare training for those who already reside rurally.

#### 4.2.2. Professional Considerations

Despite staffing difficulties in regional areas, many participants spoke of difficulties in finding rural employment. Some described obtaining work in rural areas through networking rather than through formal job application channels. They also described feeling underqualified for rural positions. Rural positions are known to require generalist skills and a flexible approach due to the larger variety of work and the fewer resources available [18]. Rural development pathways need to be available for early career health professionals to ensure that rural practice opportunities are seen as attainable and valuable by new graduates [33].

Most participants spoke of difficulties obtaining a graduate program and the competitive nature of the application process, with only one participant successfully gaining a program immediately following university graduation. Applying for a regional or rural site was seen as risky by some from urban backgrounds due to the small number of places and believing they were unlikely to be offered a place at anything other than their ‘first choice’ location due to the way the application process was set up. The fact that rural graduate programs may not have been encouraged as a valuable option for students by at least one metropolitan university suggests that a closer relationship between universities and the nursing workforce area within WA Country Health Services is required. Appropriate rural graduate positions could be developed and advertised to potential graduates with a rural interest. The availability and ease of submitting applications for rural locations may also need review, given that a longitudinal study of nursing and allied health students undertaking UDRH placements showed that ‘first job being rural’ had a significant influence on long-term rural practice [13]. Further consideration is required at a policy level to ensure systemic enhancements to enable nursing students to apply for a rural graduate program without this being detrimental to their overall likelihood of being offered a graduate program.

All three rural origin participants who completed their graduate program in a rural area had completed their undergraduate training at a rural university site. Previous research has shown that there was no difference in the rural workforce choices of students from rural backgrounds, irrespective of their university location, and proposed that both rural universities and affirmative action for selecting rural students into metropolitan nursing programs are effective workforce strategies, although rural campuses have the added benefit of encouraging under-represented rural students to access university education [41]. Access to nursing study through locally provided courses suited mature women who did not necessarily have the flexibility to move for study in more traditional metropolitan university campus-based courses. Prospectively, these rural graduates had felt that completing a graduate program in the area where they had trained would be beneficial. However, in retrospect, although they had valued the support of their fellow graduates and staff, who had been known to them throughout their rural training, two of the graduates felt that they lacked certain clinical skills compared to their metropolitan counterparts and identified that they had received ‘inconsistent’ support during their graduate program. They were dissatisfied with the professional learning opportunities available to them at the time of their final interview, with both looking at the potential of short-term metropolitan employment or secondments as opportunities to upskill. However, both were struggling to be released for these professional development opportunities due to staffing shortages. This finding warrants further investigation within a larger cohort of rurally trained nurses. It is particularly important because job satisfaction, professional support and opportunities for career development are all associated with retention in the rural workforce [9,18,30,40,42]. Conversely, a lack of professional development opportunities is a significant disincentive to the recruitment and retention of rural health professionals, so it is concerning for rural nursing workforce stability if rurally trained and rurally practising nurses are being lost to metropolitan areas because they lack professional development opportunities. Suggestions for novel professional development experiences using telehealth, clinical exchange, and service collaboration have been suggested as ways of enhancing long-term rural retention [33,43]. A previously trialled program included the opportunity to develop networks between metropolitan-based and rural-based staff who work in specific clinical areas, allowing them to share knowledge in a two-way learning experience [44]. This model has the potential to enhance networking, clinical skills, knowledge, confidence, and awareness of rural service provision [44]. Others have focused on mentorship to enhance the retention of rural and remote nurses and to overcome the issue of professional isolation [9,18,42,45]. Those in a rural position can fear becoming deskilled or diverging to a lower level of skills compared to metropolitan colleagues, and this belief could negatively influence rural healthcare worker retention, even if the true skill levels are comparable. For this reason, increasing the appreciation of the skills of rural health workers and expanding their access to ongoing professional development may be a way of enhancing their willingness to remain in a rural position for longer.

Another substantive issue raised by the participants relates to the high numbers of casual and short-term contracts in the nursing sector as it contributes to issues around the attraction and retention of a stable and sustainable rural workforce. Along with professional support and development opportunities, job security has been shown to influence the continuation of rural practice [46,47], given that short-term contracts make it harder to settle and integrate fully into a new community [47].

#### 4.2.3. Personal Considerations

Social, cultural, and family ties have been shown to influence the retention of rural health workforces [9,18,20,48]. For those without a rural background, opportunities to develop social networks in the community are particularly important[19,20,49,50]. Similarly, immersion in rural culture has been shown to be a major motivator for remaining in rural and remote communities [18]. Family considerations were mentioned by most participants with respect to making decisions about employment locations, whether they were of urban or rural origin. A partner’s work was a frequent consideration, with many staying in a certain location due to this. Partners have previously been identified as a common factor in rural employment decision-making, with pursuing a rural career being associated with being partnered with someone from a rural background [51]. This indicates that a partner’s needs have a large influence on the ability of health workers to take up rural employment, regardless of rural interest or intention, and is a key factor to be considered in rural workforce planning. This issue extends beyond nursing, and it can be hoped that more recent developments in the acceptability of work-from-home and enhanced online communication technologies could potentially positively contribute to partners having the flexibility to relocate to rural areas for work.

## 5. Summary

Table 2 below outlines the key findings of this study, along with their implications for policy and practice.

## 6. Limitations

As with most longitudinal studies, achieving interviews with the initial student participants as they moved into professional careers proved challenging. Our numbers are small but have produced rich qualitative data with a variety of viewpoints from students from each of the university nursing courses that were available in the state of Western Australia at the beginning of the study. Qualitative research does not seek to be representative, and the study participants highlight the differences within the university approaches, which warrant exploration in more depth.

Rural communities are diverse and include regional centres to very remote communities. What works regarding the attraction, recruitment and retention of rural nurses in one area may not be applicable to another. However, many of the factors identified here were not location-specific and are likely relevant to many non-metropolitan settings.

Although this six-year study elucidated the factors that most influenced these nursing graduates’ decisions around practice location, further insights might have emerged from following this cohort for longer. Of particular interest would be the follow-up of those working rurally to identify further factors that enhance or negatively impact their long-term retention in the rural workforce [41]. More detail on how the life stage of the nurses, for example, school-leavers versus mature students, influenced the factors identified would also be of interest. Despite the challenges of undertaking longitudinal studies with graduates, even small pieces of qualitative research such as this can help guide policy and the development of rural workforce strategies and programs.

## 7. Conclusions

Several factors that act to enable nurses’ rural practice intention, employment and retention have been identified, including a positive rural placement experience giving students and, subsequently, graduate nurses the confidence to be able to move away to rural areas. However, several barriers and areas for improvement were highlighted, namely issues with obtaining rural graduate employment, employment insecurity due to short-term contracts, and the concerns of rural nurses regarding professional development opportunities. Understanding these factors is essential for developing strategies for the attraction, recruitment, and retention of rural nurses and for improving the long-term stability and sustainability of the rural nursing workforce.

## Figures and Tables

**Table 1 ijerph-20-05113-t001:** Participant information and interview timing.

Participant	University	Interview 1 Year	Interview 2 Year	Interview 3 Year
A	UWA	2016	2021	-
B	USQ	2016	2018	2021
C	UND	2015	2017	2021
D	USQ	2016	2019	2021
E	UND	2016	2018	2021
F	UND	2016	2018	2021
G	ECU	2018	2019	-
H	ECU	2018	2018	2021
I	UWA	2015	2019	-
J	UND	2016	2019	2020

**Table 2 ijerph-20-05113-t002:** The key findings of this study, along with their implications for policy and practice.

Key Findings, Barriers and Enablers	Implications for Policy and Practice
Rural Intention	
Satisfaction with a rural placement was enhanced by diverse skill development, autonomy of practice, and by social and professional support as part of the clinical team and wider community.	Clinical facilitators and preceptors should focus on those factors which are known to enhance placement satisfaction and enhance confidence through providing opportunities both clinically and socially.
Quiet periods and discontinuity of preceptors adversely impacted rural placement satisfaction.	Provide professional development opportunities for rural preceptor roles which include awareness of specific issues related to students on rural placements. Increase continuity of preceptors for those on rural placement. Support from Nurse Unit Managers (NUM) in allocating preceptors to students appropriately would be of value.
Participants spoke of their rural placement as an enabler of confidence to move away from their place of origin.	Enable broader immersion in the rural ‘lifestyle’, including opportunities to ‘give things a go’ and to meet new people. Incorporate social experiences as an important element of rural placements as well as opportunities for longer placements.
Rural Employment	
Rural graduate programs did not seem to be encouraged as a valuable option for some students while at university.	Enhance relationships in rural nurse recruitment pathways, including between country health services and universities.
Some students and graduates from metropolitan areas perceived that they lacked the necessary skills and experience to work rurally.	Ensure that rural practice opportunities are seen as attainable and valuable by new graduates with information on appropriate programs imparted to nursing students at the time of their rural placement as well as through campus nurse academics.
Nurses had difficulties applying for rural graduate programs without compromising alternative graduate program opportunities.	Improve the graduate program application process to ensure that graduates can express their rural choices more freely without this being detrimental to their overall likelihood of success in the attainment of a graduate program.
Family and partner education and work considerations were mentioned by most participants with respect to making decisions about employment location, whether they were of urban or rural origin. For metropolitan-based nurses, a partner’s work was a frequently cited barrier to taking a rural position despite having an interest in working rurally. Conversely, nurses with a rural background who had been rurally trained spoke of staying in their current location due to partner and family needs.	It is important to consider the issues of family and partner needs when working to attract nursing staff to rural areas. Providing positions to suit a variety of family situations may be one solution. Rural Health West provide ‘Partner Education Grants’, which provide partners of rural healthcare practitioners with funding to train and enhance their employability in rural areas. This and other such programs could be more widely advertised to new graduates and students.
Job insecurity due to short-term and casual contracts may make it harder to relocate, settle and integrate fully into a new community.	Consider longer-term employment contracts with key stages of development to be attained to continue employment. This should include the development of assured pathways for rural nursing careers from a graduate level, and programs that encourage and support nurses who are interested in rural nursing experiences.
Rural Retention	
There was a lack of satisfaction with support and professional development opportunities in rural practice, which may impact upon retention.	Ensure that rural graduate positions offer quality support and training. Develop career pathways that ensure ongoing access to professional development and career-broadening for nurses working in rural areas.
Rural graduate nurses feared de-skilling or diverging in competency and skill compared to their metropolitan counterparts. Some expressed a desire for short-term secondments to metropolitan tertiary hospitals to develop further professional skills.	Provide enhanced professional development opportunities, including training for those interested in preceptorship and research the potential for implementing novel strategies used in rural areas interstate and overseas. These include short-term ‘swaps’ or secondments to metropolitan areas (a ‘swap’ would provide added benefits for the metropolitan nurse to experience rural nursing), online skill-sharing sessions and ‘buddy’ mentoring between rural and metropolitan nurses to discuss experiences and share knowledge through two-way learning.

## Data Availability

The data in this study may be made available by contact with the authors.

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
