# Peer review of "Tracks to Postgraduate Rural Practice: Longitudinal Qualitative Follow-Up of Nursing Students Who Undertook a Rural Placement in Western Australia"

_ijerph, 2023, doi:10.3390/ijerph20065113_

Round 1
Reviewer 1 Report
Overview and general recommendation
This is a very interesting qualitative study that assesses the ability and willingness of nurses to work in rural areas. Although it was carried out in one part of Australia, this is indeed beginning to happen in other countries. Being able to contrast views over time, as well as the variety of participants, allows for a varied approach to the issue.
Nevertheless, the manuscript has some features that need to be addressed.
Major comments
The main weakness of the manuscript is the lack of concrete data in Materials and Methods about the entire procedure carried out, as well as the lack of an adequate arrangement in the presentation of the data that appear in this section. An example of this is the reference to the fact that the interviews were recorded, but it is not specified whether they were conducted in person, by telephone...
I suggest that you use the COREQ checklist to ensure that you describe most of the methodological data of the study in this section: https://www.equator-network.org/reporting-guidelines/coreq/
Minor comments
- Page 1: Please update the template on this page, as the left column is from the year 2022.
- Line 64: There is a dot that needs to be removed just before the "[12]".
- Lines 78-82: This information should be in the Materials and Methods section.
- Lines 113, 166, 261, 309, and 586: There are double spaces on these lines.
- Table 1: Please include a column specifying the age of each participant, as opinion may vary according to this variable. This is highlighted in the Discussion.
- Table 1: Regarding participant H, his first two interviews were conducted in the same year. Is there a particular reason why he was interviewed twice in a short time?
- Lines 130-134: This information regarding the explanation of the code should be moved to Material and Methods.
- Lines 136-146: If you do not incur anonymity problems, please indicate the participants (A, B, C...) you refer to throughout this paragraph.
- Line 156: To maintain the format of the template, please consider using subsection "3.2.1.A" to indicate this subheading, as well as the others in Results.
- Lines 193-194: It seems that many of the participants performed nursing activities in hospitals, irrespective of rural or urban settings. Were there any responses concerning primary care?
- Lines 277-279: Following on from the previous comment, there seems to be a stigma attached to the usefulness of primary care, when it is vital for proper health promotion in the population. This is information that is not covered as much in the manuscript but could be of interest to the study.
- Lines 314-318 and 414-416: Please correct the spacing.
- Lines 388 and 399-400: This supplementary information on location concealment should be explained in Material and Methods and not in Results.
- Line 451: Remove the square brackets that exist in citation 28.
- Line 480: There is a typo in "provsion". From here, there are several similar typos in the rest of the text. Please check this.
- Line 699: Delete "Sep 28".
- Lines 756 and 763: Please remove the comma after "Collegian".
I hope my comments will help you to improve the manuscript.
Best regards.
Author Response
Major comments
Point 1: The main weakness of the manuscript is the lack of concrete data in Materials and Methods about the entire procedure carried out, as well as the lack of an adequate arrangement in the presentation of the data that appear in this section. An example of this is the reference to the fact that the interviews were recorded, but it is not specified whether they were conducted in person, by telephone...
I suggest that you use the COREQ checklist to ensure that you describe most of the methodological data of the study in this section: https://www.equator-network.org/reporting-guidelines/coreq/
Response 1: Thank you for this suggestion. The Materials and Methods section has now been thoroughly updated using the COREQ checklist.
Minor comments
Point 2: Page 1: Please update the template on this page, as the left column is from the year 2022.
Response 2: This has been changed to 2023.
Point 3: Line 64: There is a dot that needs to be removed just before the "[12]".
Response 3: This has been removed.
Point 4: Lines 78-82: This information should be in the Materials and Methods section.
Response 4: This information has now been moved to the Materials and Methods section.
Point 5: Lines 113, 166, 261, 309, and 586: There are double spaces on these lines.
Response 5: This has been reviewed and is now consistent with the rest of the article text.
Point 6: Table 1: Please include a column specifying the age of each participant, as opinion may vary according to this variable. This is highlighted in the Discussion.
Response 6: Thank you for this suggestion. The age of the participants was originally included in this table. However, after some discussions we have removed the specific ages of the participants. This was due to an anonymity concern. In particular there are only a small number of regional graduates working in the regional hospital and, along with other personal information contained within some of the quotes and text, we felt that these persons could potentially be identified if we were to provide their ages.
Point 7: Table 1: Regarding participant H, his first two interviews were conducted in the same year. Is there a particular reason why he was interviewed twice in a short time?
Response 7: Thank you for pointing this out. This has been checked and the interviews took place in February and December 2018, therefore giving 10 months approximately between the two interviews. The gap between interviews was dependant upon timing of the previous interview and both participant and staff availability.
Point 8: Lines 130-134: This information regarding the explanation of the code should be moved to Material and Methods.
Response 8: This information has now been moved to the Materials and Methods section.
Point 9: Lines 136-146: If you do not incur anonymity problems, please indicate the participants (A, B, C...) you refer to throughout this paragraph.
Response 9: This paragraph has now been updated to incorporate this suggestion.
Point 10: Line 156: To maintain the format of the template, please consider using subsection "3.2.1.A" to indicate this subheading, as well as the others in Results.
Response 10: Thank you for this suggestion. This format change was trialled but it did not seem to flow well with the rest of the text so it has been left in its original format.
Point 11: Lines 193-194: It seems that many of the participants performed nursing activities in hospitals, irrespective of rural or urban settings. Were there any responses concerning primary care?
Response 11: Unfortunately there were no substantial perspectives on primary care arising in the analysis. One participant spoke of a primary care rural placement but there was nothing of note that we could add around primary care from the qualitative analysis. This would however be a very interesting area for further study.
Point 12: Lines 277-279: Following on from the previous comment, there seems to be a stigma attached to the usefulness of primary care, when it is vital for proper health promotion in the population. This is information that is not covered as much in the manuscript but could be of interest to the study.
Response 12: As per Response 11, we agree that this would be very interesting and that primary care nursing is vitally important for population health, but unfortunately this was not a perspective that arose during our interviews.
Point 13: - Lines 314-318 and 414-416: Please correct the spacing.
Response 13: These lines have now been reformatted to be consistent with the rest of the text.
Point 14: Lines 388 and 399-400: This supplementary information on location concealment should be explained in Material and Methods and not in Results.
Response 14: This information has now been moved to Material and Methods and removed from the Results section.
Point 15: Line 451: Remove the square brackets that exist in citation 28.
Response 15: The square brackets have been removed.
Point 16: Line 480: There is a typo in "provsion". From here, there are several similar typos in the rest of the text. Please check this.
Response 16: Thank you – this has been altered to the correct spelling and the rest of the text has been re-checked.
Point 17: Line 699: Delete "Sep 28".
Response 17: This has been deleted.
Point 18: Lines 756 and 763: Please remove the comma after “Collegian”.
Response 18: These commas have been removed.
Point 19: I hope my comments will help you to improve the manuscript.
Response 19: Thank you very much for your very thorough suggestions and comments – they have been extremely helpful in refining the manuscript.
Reviewer 2 Report
Congratulations to the authors for conducting such a comprehensive study, with a longitudinal qualitative approach. I propose some aspects for improvement, in order to achieve publication in the journal:
Introduction: Incorporate some background information about rural nursing and previous research that would allow us to know, in greater depth, the knowledge gap that this research fills.
Methodology:
Has information saturation been reached?
Has any triangulation strategy been employed in the information analysis process?
Discussion/conclusions: What implications does the research have for practice.
Provide further conclusions from the information in the results.
Author Response
Point 1: Introduction: Incorporate some background information about rural nursing and previous research that would allow us to know, in greater depth, the knowledge gap that this research fills.
Response 1: Thank you for this suggestion. We have added further information into the Introduction (Lines 79-95) regarding previous relevant research and knowledge gaps. We have also expanded on how this study aimed to fill these gaps (Lines 97-110).
Methodology:
Point 2: Has information saturation been reached?
Response 2: This information has now been added in Lines 177-178.
Point 3: Has any triangulation strategy been employed in the information analysis process?
Response 3: This information has now been added in Lines 179-183.
Discussion/conclusions:
Point 4: What implications does the research have for practice.
Response 4: Table 2, second column, outlines the implications of this research for practice in this field. This has now been made clearer.
Point 5: Provide further conclusions from the information in the results.
Response 5: We have provided conclusions from the results to the best of our knowledge and could not find any further conclusions to add at this time. If there was a particular aspect that you would like us to further address or comment upon then we would be happy to revisit this area and provide further discussion. Thank you very much for your review and assistance with refining this manuscript.
Reviewer 3 Report
<Introduction>- As suggested in the first sentence, it will be a universal opinion that nurses are the main medical personnel, but evidence is needed to assert this. Please provide a reference.
- In a situation where there is only an explanation for the anticipated nurse shortage and no explanation for the reality of Australia's rural health workforce, suddenly there is an explanation that nurses are the backbone of the rural health workforce. The research question must be clearly stated. A clear description of the current state and anticipated challenges of the rural health and nursing workforce is needed.
- I think the introduction about CF is unnecessarily long. It is also not considered a necessary description in this study.
<Data collection and analysis>
- Please explain why the number of interviews with study participants differed from 2 to 3.
- It is desirable to describe the research problem so that follow-up research is possible.
Author Response
<Introduction>
Point 1: As suggested in the first sentence, it will be a universal opinion that nurses are the main medical personnel, but evidence is needed to assert this. Please provide a reference.
Response 1: Thank you for pointing this out. This reference has now been added in Line 32.
Point 2: In a situation where there is only an explanation for the anticipated nurse shortage and no explanation for the reality of Australia's rural health workforce, suddenly there is an explanation that nurses are the backbone of the rural health workforce. The research question must be clearly stated. A clear description of the current state and anticipated challenges of the rural health and nursing workforce is needed.
Response 2: Thank you for this suggestion. This paragraph (Lines 27-40) has now been updated to give a clearer description of the issues and how this is likely to impact upon health care provision in rural areas.
Point 3: I think the introduction about CF is unnecessarily long. It is also not considered a necessary description in this study.
Response 3: Thank you for this suggestion. This paragraph has been modified and shortened to give a more concise explanation (Lines 73-78).
<Data collection and analysis>
Point 4: Please explain why the number of interviews with study participants differed from 2 to 3.
Response 4: This information has now been added in Lines 166-168.
Point 5: It is desirable to describe the research problem so that follow-up research is possible.
Response 5: Thank you for this suggestion. This information has now been updated and expanded upon in Lines 97-110.
Round 2
Reviewer 3 Report
Modification recommendations have been generally reflected.